# The Characteristics of the Religiosity of Youth in Slovakia Today

Peter Kondrla * and Eva Ďurková

Institute for the Research of Constantine and Methodius' Cultural Heritage, Faculty of Arts,
Constantine the Philosopher University, 949 01 Nitra, Slovakia; durkova.eva15@gmail.com
* Correspondence: pkondrla@ukf.sk

**Abstract:** This paper discusses the current characteristics of the religiosity of young people in Slovakia based on data that were obtained in a survey as part of the research project "Perspectives of the Development of Religiosity in Slovakia", carried out in 2017–2020. The data presented in the article pertain to several areas related to the religious lives of young people. Firstly, data on general changes in the respondents' religiosity (such as a weakening or a strengthening) are presented, followed by data on the causes that the respondents consider having influenced these changes. Secondly, since many of the respondents claimed factors related to receiving a lot of information as such causes, the article presents data to determine to what extent the respondents paid attention to religious media in Slovakia (newspaper, radio, television, and websites). Lastly, we present the data on the respondents' attitudes towards the moral acceptance of some themes that are religiously controversial or politically divisive in Slovak society today. We then discuss the indications that emerge from the data on the religiosity of young people in Slovakia today, overall concluding that there is often an occurrence among Slovak youth of claiming an affiliation to an official church/confession/religious institution while developing and acting on individual religious opinions or beliefs.

**Keywords:** religiosity; changes; opinions; religious media; moral acceptance; youth; Slovakia

## 1. Introduction

Between 2017 and 2020, the authors of this article and their colleagues worked on a research project named "Perspectives of the Development of Religiosity in Slovakia." The project was funded by the Slovak APVV Agency (Slovak Research and Development Agency) and focused on detecting changes in the religious life of the Slovak population. Within the project's research, a survey aimed to detect how such changes came about. A questionnaire providing respondents with a set of questions related to their religious life and religious opinions was distributed using the services of the data collecting company Data Collect s. r. o. The research sample included a total of 1000 respondents. In the questionnaire, 58% of the respondents claimed that there had been no changes in their religiosity today compared to the past or were unable to answer, while 42% claimed that their religious life had changed (in the form of either a weakening or a strengthening) compared to the past. The percentage of those who experienced a change was even higher among respondents from 18 to 24 years of age, on whom we will focus in this article and, from now on, will also refer to as 'young respondents'. Among the young respondents, 57.2% of them claimed such a change, while 42.8% declared that their religiosity had not undergone any changes or chose the option "hard to tell." The young respondents represented 124 out of the total 1000 respondents from the research sample. The questionnaire consisted of 30 questions dealing with various religion-related themes. In this article, we will present the data obtained from the young respondents on several of these survey questions, firstly, the data related to already-introduced changes in their religiosity over time; secondly, the data related to their engagement in following religious mass media in Slovakia; and

thirdly, the data related to the respondents' moral attitudes towards some controversial and society-dividing religious topics/issues in Slovakia today. Afterward, we will discuss the conclusions implied in the obtained data and reflect on their possible indications regarding Slovak youth and religion. The theorem of this paper is that there is discordance between the teachings of the official confessions and religious institutions that most young people in Slovakia claim to belong to and the characteristics of their actual religiosity and beliefs. We back up the theorem with data that will be presented herein.

## 2. Results

As we have already mentioned, the sample included 124 respondents of 18–24 years of age, also referred to as young respondents in this article. Before presenting the data on the questions introduced above, we will provide the data on some general religion-related characteristics of these respondents as obtained through the questionnaire. Regarding the official church/confession/religious institution that the young respondents do or do not belong to, 73.3% of them claimed an affiliation with the Catholic Church (either Roman or Greek Catholic), 6.5% with the Evangelical Church, 15.3% with no church or religious institution, and the rest identified with several other churches (overall, the data on these characteristics fit the data of the overall population). Regarding the high percentage of Christians, we might clarify that Christianity in Slovakia played an important role both during the national revival in the 19th century and during the communist regime, when Christianity was often persecuted and restricted (Vargová 2021). At present, Christianity in Slovakia manifests itself not only as a religion but also as a factor of culture and identity. Christian symbols and customs are found in various areas of Slovak society, such as holidays, traditions, and cultural events (Judák 2022, p. 516). Continuing our presentation of the general religious characteristics of our respondents, 85.5% of them claimed to not belong to any other religious community beyond the official churches/religious institutions, 6.5% claimed that they do belong to such a community and are active in that community, and 8.1% said they belong but are not active. Finally, regarding the actual state of their inner life when it comes to faith, beyond church affiliations or teachings and regarding only the respondents' faith itself, 55.6% of the young respondents described themselves as believers or deep believers, 21% as non-believers, 4% as unconcerned about faith, and 19.4% as undecided but tied to a religious tradition.

Turning our attention to the focus of this article as described in the introduction, we will now present the data we obtained when the respondents were asked to choose the option that best fit their understanding of the developments in their religious lives over time, i.e., if they were less or more religious today compared to the past. Figure 1 shows that 30.6% of the respondents thought that there had been no changes, 12.1% could not answer satisfactorily, 38.7% expressed that they were more religious in the past, and 18.5% considered themselves more religious today than in the past.

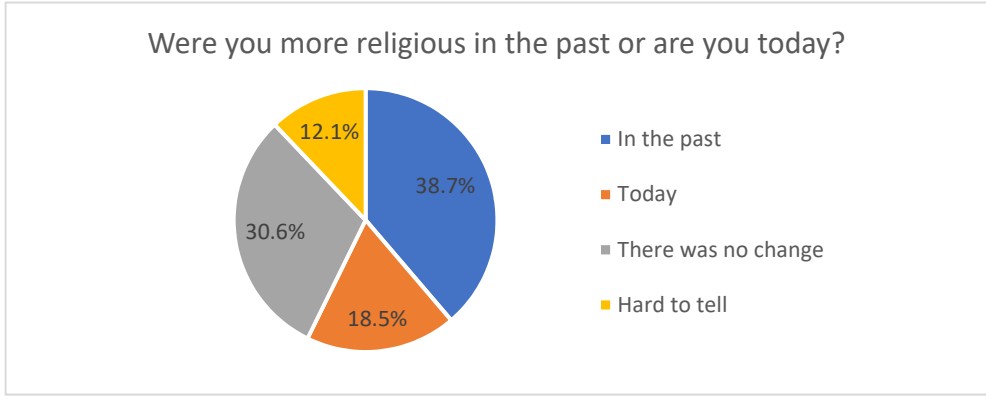

**Figure 1.** Changes in the religiosity of respondents comparing today with the past.

Next, those who declared a change in their religiosity, either in the form of a weakening or strengthening, were asked to provide information on what factors they considered to have influenced that change. They were given a wide set of options to provide this information, such as "Church, its behavior and scandals"; "Information, personal outlook, and experiences"; "Growing up, developing individual opinions"; "Life events (like death, illness, hardships)"; "Dogmatic upbringing"; "Love, well-being, peace"; "Personal experience, miracles, healings"; "Better understanding God, knowing God"; "COVID period and COVID-related restrictions"; and several others, including the options "Other" and "I do not know/none." The respondents could choose one or an unlimited number of options provided. The additional option to obtain the details from those who chose the option "other" was not provided.

Figure 2 shows that among those who expressed that they were more religious in the past, the highest percentage chose the option "I do not know/none" (20.8%). Next, "Church, its behavior, scandals" (18.8%) and "Information, personal outlook, and experiences" (18.8%) were significant, followed by "Growing up, developing individual opinions" (10.4%) and "Dogmatic upbringing" (8.3%). In addition, 12.5% of young respondents chose the option "Other".

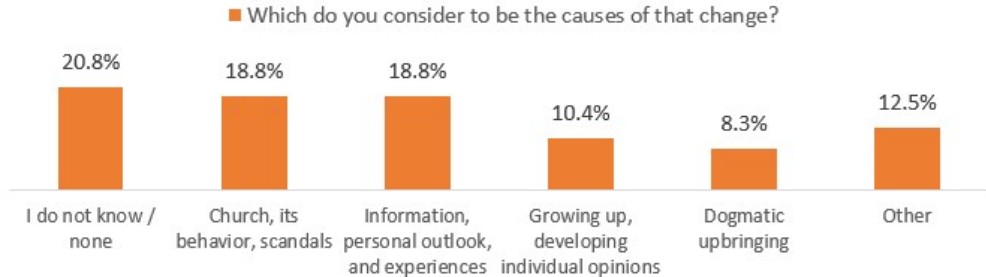

**Figure 2.** Factors that those who were more religious in the past consider as aiding in the change in their religiosity.

As Figure 3 shows, among respondents who declared being more religious today than they were in the past, 17.4% chose the option "Love, well-being, peace", 17.4% chose the option "Personal experience, miracles, healings", 13% chose the option "Life events (like death, illness, hardships)", and 8.7% chose the option "Better understanding God, knowing God". As for those who could not answer, 8.7% chose the option "Other" and 13% the option "I do not know/none".

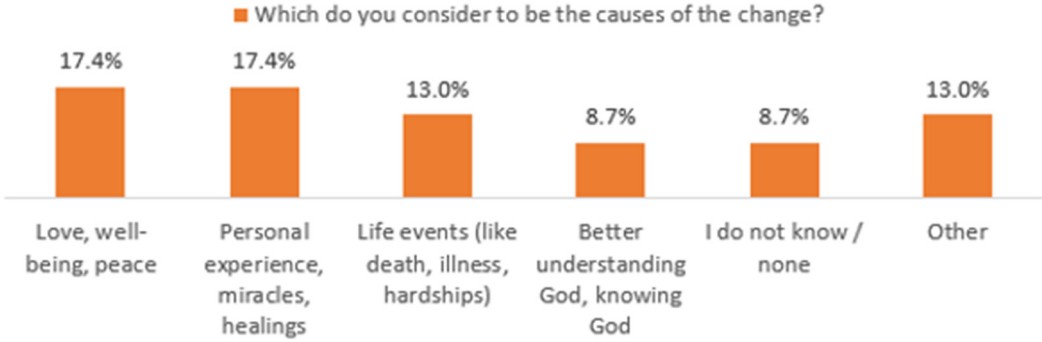

**Figure 3.** Factors that those who are more religious today consider as aiding in the change in their religiosity.

As 18.8% of young respondents who were more religious in the past consider "Information, personal outlook, and experiences" one of the causes of weakening their religiosity, it is reasonable to suggest that a rational way of processing information plays a prominent role for a significant number of young people in Slovakia nowadays when it comes to their religiosity. Young adults from 18 to 24 years of age, sometimes also called adolescents, rep-

resent a group characterized by certain generally known characteristics. They are situated in a dynamic life period. They are often students, recent school graduates, or first-time employees, undergoing change in their social environment as they enter a new study program or begin their professional life. They classify as members of the generation known as 'Generation Z', characterized by the fluent usage of the internet (since they started to use it in childhood) and access to various sources of information. Thus, they possess the ability to process information in various ways. Value orientations are important in their personal decisions and critical thinking plays an important role in their formation (Azizi et al. 2023). Diverse information constantly reaches members of Generation Z, and the factors that influence how they process this information to form opinions and develop values include their social group, as well as the emotional transformation that is characteristic of this biological age. As Jarmoch et al. (2022, p. 140) state, part of creating a view of the world is the search for the meaning of existence as well as the search for a place in a social group. Such complexly situated young people also reflect on religious and moral issues. The data obtained from those who were more religious in the past seem to sustain that.

The fact that 18.8% of those who were more religious in the past identified "Church, its behavior, and scandals" as one of the causes for the weakening of their religiosity, shows firstly that many young people perceive the Church as a social group to which they indeed belong, and upon which they apply their reasoning and critical thinking. Secondly, it shows that the discredit of the Church affects the depth of the religiosity they share with that social group. The data thus imply that the discord between the moral norms promoted by the churches and the scandalous actions of their representatives that go against those norms might trigger critical thinking and rational reflection applied upon the Church by young respondents.

On the other hand, the causes of the strengthening of their religious life chosen by those who declared to be more religious today seem to be more intensively linked to emotionality and less intensively to rationality compared to the previous group (i.e., those who were more religious in the past). The options "Love, well-being, peace" and "Personal experience, miracles, healings", both chosen by 17.4% of young respondents, represent a highly subjective response, and the interpretation of miracles and healings is rather complicated. Since it is difficult to prove their relevance or real connection to God or transcendence, and they go beyond the limits of fact-based rational and critical thinking, the responses suggest that for the significant number of young people in Slovakia, emotionality plays a prominent role in the strengthening of their religious life.

The obtained data thus Indicate that those who prefer rational processing of Information are more likely to not only abandon the traditional beliefs but also weaken their religiosity, and those who draw information from sources of inner emotional life are more likely to stick with and strengthen their religiosity.

As respondents who reported being more religious in the past than today often claimed the behavior of the Church and their personal outlook as the causes, and both causes are information-based, questions might arise regarding where the respondents draw their information from and how they process it. It is generally agreed that when making decisions and forming opinions based on information, humans rely heavily on the emotional dimension, along with the content of the information itself. Of course, for Generation Z members, the information commonly comes from media, which is often designed to evoke an expected atmosphere and emotional state, or to influence the recipient's decision (Valčová et al. 2021); however, for most young people, the emotional dimension is usually mostly impacted by their environment of friends, family, religious and other social groups, etc. Still, information coming from the media is a common phenomenon of the digital age in which Generation Z youth live, and it touches the religious context as well. According to Štefaňák, who carried out long-term research focused on the religiosity of Slovak youth in northern Slovakia, media communication influences young people's decision-making in the area of religiosity (Štefaňak 2020a). This could be especially true for digital media. According to Campbell and Tsuria (2022), religion and digital media have

a complex relationship that can be positive, negative, or balanced. In their book *Digital Religion: Understanding Religious Practice in Digital Media*, they identify several trends in this relationship. Religious institutions use digital media to spread their message and recruit new believers. This can increase the availability of information about religion and improve interaction between believers and religious leaders. Digital media can facilitate the emergence of new religious communities and alternative religious groups not associated with traditional institutions. Digital media can lead to the individualization of religious practices and opinions because it allows people to access a variety of different data and opinions and form their own opinions and beliefs. At the same time, however, there is also a risk that digital media can lead to misinformation and the emergence of extremist views and attitudes that may conflict with religious values. In some cases, the use of digital media can even replace traditional religious experiences and rituals, such as prayer, worship, and communal religious observances.

While in our research we did not probe into where exactly the respondents acquired their information from, we wanted to know the position of religious media in the context of delivering religious-related information to young people in Slovakia, whether the religious media channels in Slovakia satisfy the same trend in importance as the secular media, and whether they are commonly followed by young people. The young respondents were therefore asked to claim the extent to which they follow the religious media in Slovakia. The questions focused on religious magazines and newspapers, religious radio, religious television, and religious websites. Additionally, because young adults are generally expected to be more eager to follow news and the information coming from mass media compared to when they were children, with each of the mentioned types of media, we asked respondents to report the extent to which they followed it ten years ago as well as the extent to which they follow it today. Since, as stated earlier, 55.6% of young respondents self-identified as deeply believing or believing, and 79.8% of young respondents claimed affiliation with one of the Christian churches in Slovakia (Roman Catholic, Greek Catholic, or Evangelical), we expected a significant number of young respondents to follow the given religious media sources at least occasionally.

**Religious newspapers and magazines**. Figure 4 shows the data obtained regarding the questions "How often do you read religious newspapers and magazines?" and "How often did you read religious newspapers and magazines approximately ten years ago?". The respondents could choose from the options "Every number", "Almost every number", "Occasionally", "Rarely", and "Not at all".

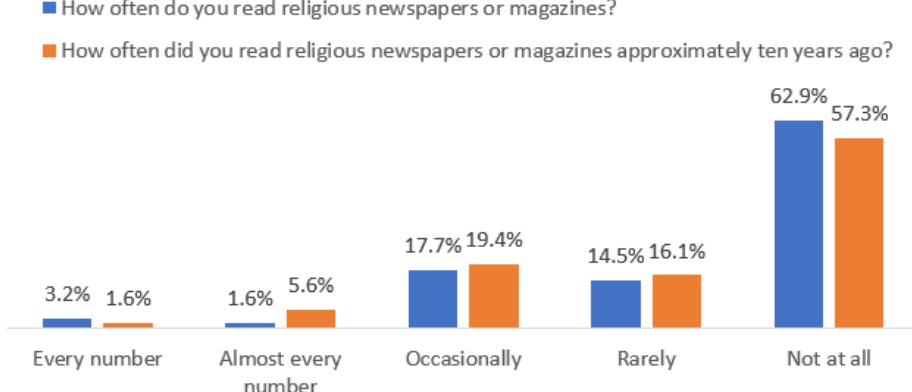

**Figure 4.** The respondents' declarations on the frequency of reading religious newspapers and magazines today and approximately ten years ago.

As shown in Figure 4, most young respondents do not read religious newspapers or magazines today, and most of them did not read them ten years ago as well. More specific research, however, would need to be conducted to find out how many 'occasional' readers or 'not at all' readers are believers and how many are non-believers. The questionnaire did not cover these specifics since it was not one of the main goals of the original and complex

30-question survey. It also needs to be stated that the question did not specify whether it referred to the print or online newspapers. However, the respondents could claim visiting any online religious websites in another question, which we will deal with in a moment.

**Religious radio**. Figure 5 shows the data obtained regarding the questions "How often do you listen to religious radio?" and "How often did you listen to religious radio approximately ten years ago?". The respondents could choose from the options "Daily", "Several times a week", "Several times a month", "Several times a year", and "Not at all".

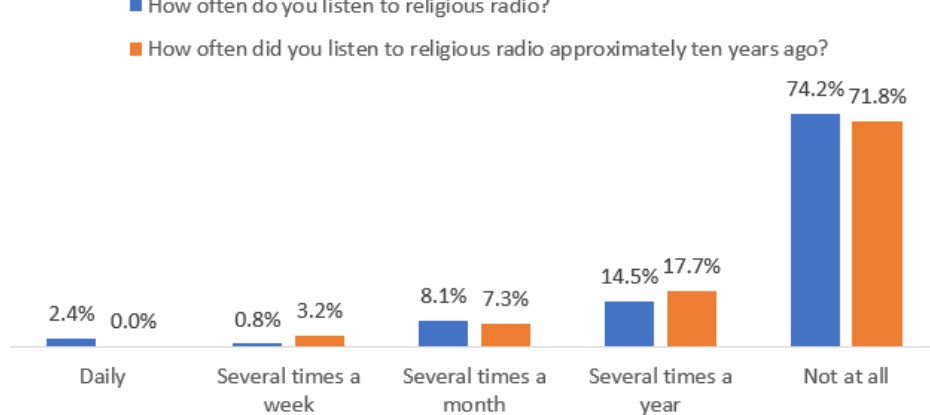

**Figure 5.** The respondents' declarations on the frequency of listening to religious radio today and approximately ten years ago.

As shown in Figure 5, the percentage of those who stated they do not listen to religious radio was even higher than in the case of newspapers and magazines. According to this trend, the percentage of those who listen to it at least occasionally is also much lower than in the previous figure.

Religious television. Figure 6 shows the data obtained regarding the questions "How often do you watch religious television?" and "How often did you watch religious television approximately ten years ago?" The same options were applied as presented in Figure 5.

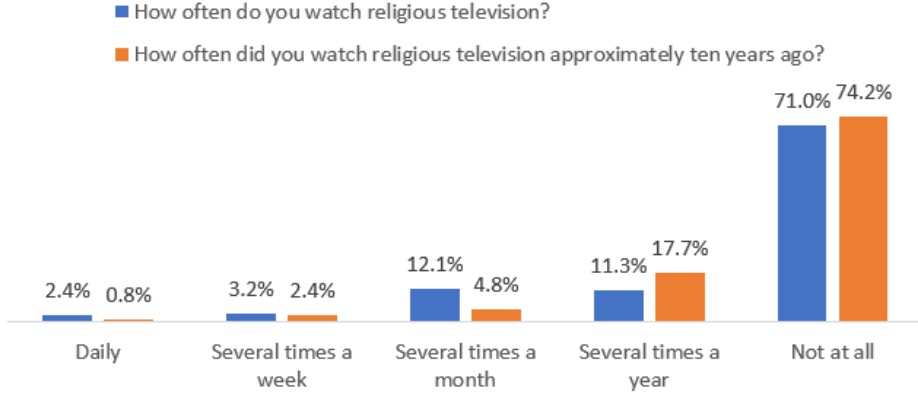

**Figure 6.** The respondents' declarations on the frequency of watching religious television today and approximately ten years ago.

Figure 6 shows similar results related to religious television as Figure 5, which dealt with religious radio. In the case of religious television, however, the percentage of those who stated they do not watch it today was slightly, although not statistically significantly, lower than the percentage of those who did not watch it ten years ago.

Religious websites. Figure 7 shows the data obtained regarding the questions "How often do you visit religious websites?" and "How often did you visit religious websites approximately ten years ago?" The same options were applied as presented in Figures 5 and 6.

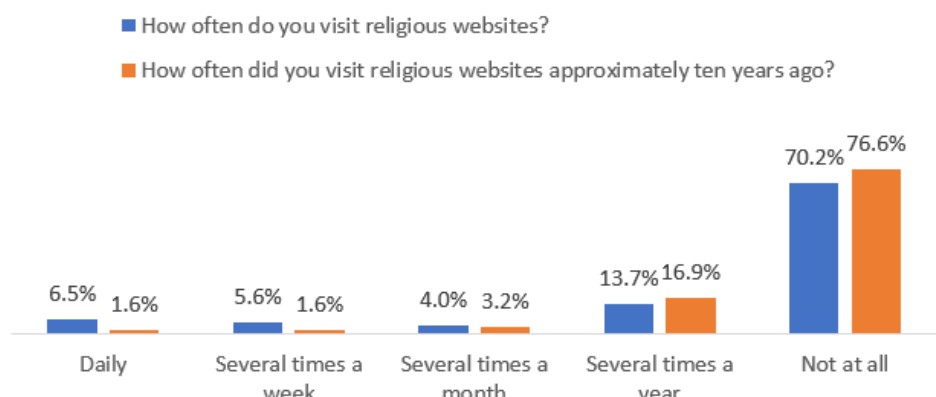

**Figure 7.** The respondents' declarations on the frequency of visiting religious websites today and approximately ten years ago.

Although we expected the percentage of those who visit religious websites to be the highest of the four given types of religious media, it turned out not to be the case. As we can see in Figure 7, the data are similar to those found in the case of religious television.

The data produced by these questions about religious media make sense in the context of the data presented earlier in this article, since the information that most of Slovak youth do not follow religious media, and most of the rest only follow it occasionally, means that they probably do not use religious media to seek information and opinions on church scandals or to aid in forming their personal outlook. In Slovakia today, whether it is on the level of national media or the level of local religious communities, scandals related to the Church most often involve the engagement of clergy (and other Church-protected representatives) in sexual activities. In addition, there are other topics/issues that also have the potential to discredit the Church in the eyes of its members and impact their religiosity. For these, we consider firstly the topics/issues that are religiously controversial and politically divisive in Slovakia today: induced abortion and registered partnership for same-sex couples (not legalized in Slovakia to date). Secondly, there are the issues related to the fact that most people, whether believers or non-believers, engage in sexual behaviors which are condemned by the largest churches' teachings in Slovakia. Most prominent examples would be premarital sexual relationships and the use of contraceptives, either premarital or while married. These issues can be problematic for young people, possibly impacting their relationship to Church and their religiosity. Our questionnaire was not specifically designed to confront respondents with questions about such impacts; however, it was able to determine the extent to which young respondents morally accept these issues. In Figure 8a–d, we show the data we obtained from young respondents regarding their moral acceptance/condemnation of premarital sexual relationships, the use of contraceptives, homosexuality, and induced abortion. The respondents were choosing from the options "I consider it morally approved", "I consider it morally approved in certain circumstances", "I consider it morally disapproved", and "Hard to tell". The respondents were again asked about their attitudes approximately ten years ago as well. Let us first present all the figures and provide the description and commentary afterward.

The figures show that in all cases except for the case of induced abortion, at least 50% of respondents fully morally approve of the given issue. Since 73.3% of young respondents in our survey confirmed their affiliation with the Roman or Greek Catholic Church (and others with the Evangelical Church), this shows the dissonance between respondents' moral settings and the teachings of their churches. Although 79.8% of respondents stated their affiliation with these Christian churches, only 18% on average said they morally disapproved of homosexuality, the use of contraceptives, or premarital sexual relationships. Additionally, when we look at the percentage of respondents who morally approved of these three behaviors ten years ago, it shows there is a trend of young people becoming more morally tolerant toward these issues compared to their childhood. Of all four issues

that we confronted the respondents about, only induced abortion showed an approximate correlation with the data on the respondents' affiliation with the Christian churches. A total of 73.3% of respondents stated their affiliation with the Roman or Greek Catholic Church, and 62.1% of them said they either morally disapprove of induced abortion or morally approve of it only in certain circumstances, as does the Catholic Church's moral theology in certain rare circumstances.

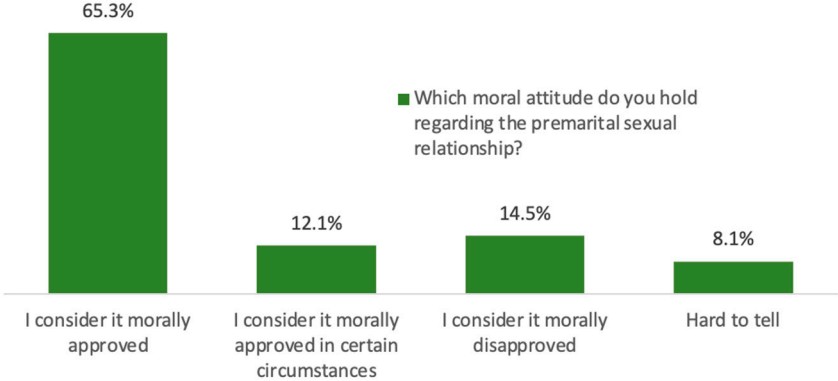

50.8% of respondents declared, that they considered premarital sexual relationship morally approved approximately 10 years ago.

(**a**)

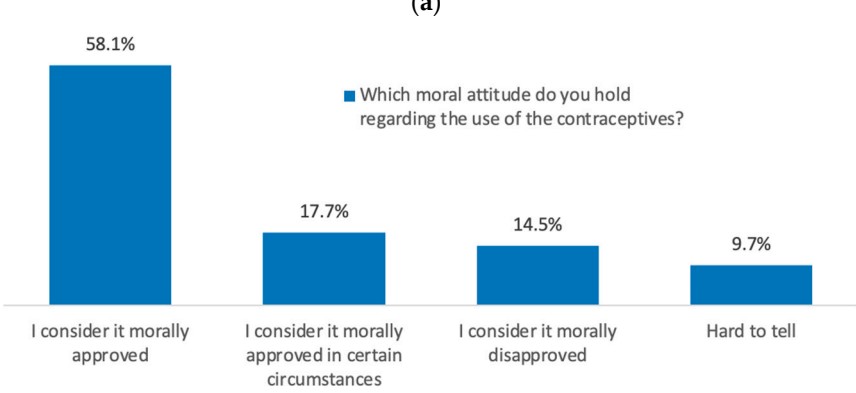

46.0% of respondents declared that they considered the use of contraceptives morally approved approximately ten years ago.

(**b**)

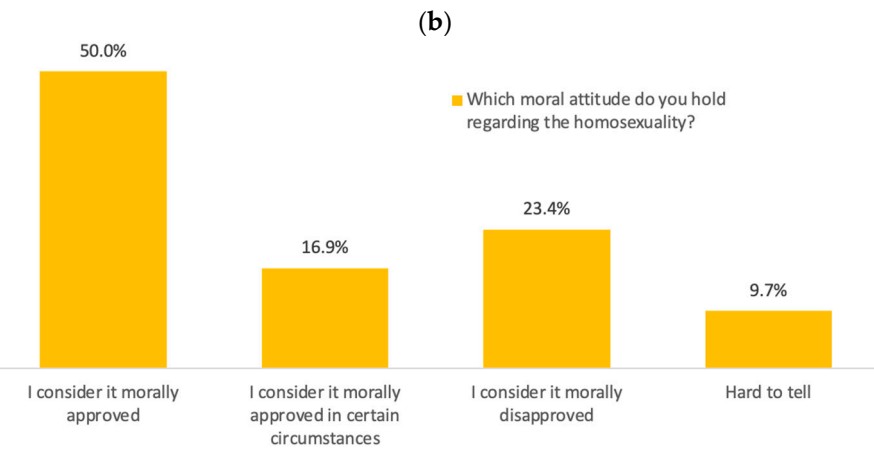

36.3% of respondents declared that they considered homosexuality morally approved approximately ten years ago.

(**c**)

**Figure 8.** *Cont.*

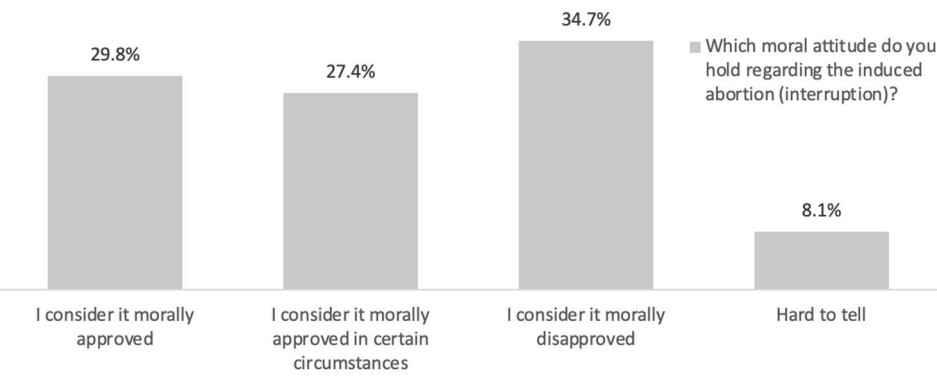

21.0% of respondents declared that they considered induced abortion morally approved approximately ten years ago.

(**d**)

**Figure 8.** (**a**) Respondents' moral attitude toward premarital sexual relationships. (**b**) Respondents' moral attitude toward the use of contraceptives. (**c**) Respondents' moral attitude toward homosexuality. (**d**) Respondents' moral attitude toward induced abortion.

## 3. Discussion

In this article, we provide data related to several aspects of the religiosity of young Slovaks: changes in their religiosity over time and the factors they consider to have impacted those changes; the influence of the religious media of Slovakia on Slovak youth; and the moral attitudes of young Slovaks towards several religiously controversial topics in Slovakia today. Here we discuss the indications and interrelationships emerging from these data.

We found that there were more than twice (over 38% of the total) as many young respondents who were more religious in the past compared to today, than those who are more religious today (over 18% of the total) compared to past. Of those who were more religious in the past, 18% considered the behavior and the scandals of the Church as factors aiding in weakening their religious life, and another 18% attributed the weakening to changes in personal outlook and experiences. Since both causes are very much information-based, they can be classified as more rationally framed than emotionally framed. On the other hand, the causes most frequently considered as aiding in strengthening respondents' religiosity were love, well-being, and peace and personal experience, miracles, and healing (both chosen by more than 17% of respondents). These reasons are characterized by more of an emotional, rather than rational, nature. Together, these different kinds of reported causes represent the general fact that when processing information and making decisions, the information itself and the emotion related to it are both important to human beings. It matters very much whether the information makes sense to a person, and the value of information also largely depends on how it hits the emotional component of a person (Kobylarek et al. 2022; Hlad et al. 2022). Young people belonging to Generation Z, who are used to being recipients of much diverse information since their childhood, live in an age in which one of the powers able to provide both the information itself and the emotion embedded into it, is the media (Hossain et al. 2022). The media as known by Generation Z is highly influential in shaping not only judgments but also values and both individual and collective decisions (Čergeťová Tomanová et al. 2021). For young adults, the need to follow the news and current happenings related to their lives is greater compared to when they were childhood. For religious media, however, we found this not to be the case. We expected that most young respondents would not follow religious newspapers, radio, television, or websites ten years ago when these respondents were children of lower or higher secondary school age; however, we did not expect that the vast majority, 70% of respondents in average, do not pay attention to it today. Given that 55.6% of young respondents in our survey identified as believing or deeply believing, that 67.7% claimed

affiliation with the Roman Catholic Church (and some others with the Greek Catholic and Evangelical churches), and that young people are fluent in engaging with the digital world, it was surprising that only 14.3% of them, on average, might look for religious-related information in the religious media at least occasionally or several times a year. However, not an insignificant number of respondents who were more religious in the past stated that the causes for this were personal outlook and the Church's behavior; therefore, they must be consuming religion-related information somewhere. The reality that most young adults in Slovakia do not pay any attention to religious media is an incentive for further research on the causes of that reality and a potentially useful finding for churches in Slovakia that produce media that could help them in defining and adapting their communication strategies towards young people, especially in today's society described by some as a society where the media has become like a neural network that transmits information and news from near and far corners of the earth and beyond (Budayová et al. 2022). Since young people who were more religious in the past generally claimed the importance of information and personal outlook, they might be more likely to trust the media reports (Tkáčová et al. 2023). On the other hand, those who are more religious today and generally indicated more of an importance of emotional information (coming from healings, miracles, and inner life experiences) might be, in a way, protected from the inflow of external information by their conservative approach; they might experience a certain isolation, which prevents them from more comprehensively navigating the issues during our rapidly changing times (Tkáčová et al. 2022), or they might seek emotional information in the content of religious media. Based on the significance of personal experiences in their inner life, this suggests that young believers appreciate a conservative approach where they cultivate their values through experiences of historical tradition or their personal history (Rychnová et al. 2022). In any case, it would be interesting for researchers, as well as churches, to see if and which media sources, in general, young people pay attention to and how regularly. Today, the media serves several purposes, from education, upbringing, and entertainment to manipulation, abuse, and harm (Gawroński et al. 2021), and surely it does not distribute religious-life-related information. The media saturation of our society has an impact on the young generation, sometimes tracked by the time young people spend in the presence of media content (Kobylarek 2019). Of course, there are those in Slovakia who underline the negative impact of the media on young people or society, suggesting that various media create an environment devoid of critical thinking (Nemec 2021), or even that "The media world distorts the view of events that belong to the common experience in society" (Akimjak et al. 2022, p. 47). Critical thinking, of course, manifests in various ways of communicating, the disposition for constructive problem solving, and the overall quality of the social environment (Kondrla et al. 2023), and indeed, as a part of the school system, the subject of media education is already introduced into the curriculum in Slovakia (although it cannot yet be verified to what extent this subject fulfills its purpose) (Khonamri et al. 2022). Generally, the positive effects of media certainly exceed its negative impacts, and it acts as a valuable space for the storage of information for various purposes. In every case, our survey confirmed that for young people in Slovakia, the religious media generally does not occupy this space. As for our research findings, a dissonance was detected between the number of young respondents who declared Church affiliation or being believers or deep believers and the number of respondents who follow the religious media in Slovakia. Thus, more research attention should be paid to how exactly young people work with the media regarding their religious attitudes. While young people are free to accept or reject the informative value of media content, there are probably those who do not pay attention to the reliability of the source and prefer sensational news to verified news (Čaja 2022), as well as those for whom the media functions as a tool for affirming their already-formed beliefs (Lešková and Ďatelinka 2019).

Another finding from our research is that young people are sensitive to the dissonance between the official teachings of the churches and the scandalous behavior of their representatives that goes against those teachings, even to the point where such discredit of the Church aids in weakening their religiosity. Could it be that they are also sensitive towards

the dissonance between the biggest churches' teachings on certain forms of behavior and the fact that people claiming to be members of those churches commonly engage in such behavior? The findings from our survey showing that over half of young respondents morally accept premarital sexual relationships, the use of contraceptives, and homosexuality, and at least 30% of them morally accepted induced abortion, certainly beg such a question. As a reminder, almost 80% of our respondents claimed affiliation with one of the biggest churches in Slovakia. Except for induced abortion, which the Catholic Church's moral theology approves of in certain rare circumstances, these churches disapprove of premarital sexual relationships, the use of contraceptives, and homosexuality. Yet, of all the young respondents, only 18%, on average, morally disapproved of (or approved of only in certain circumstances) these behaviors. Induced abortion, which 35% of respondents morally disapproved of, and 27% approved of in certain circumstances, was shown to be the only behavior among the given cases where respondents' responses appeared to be in coherence with data on their faith and church affiliation. Beyond the fact that the data also suggested that there is an increasing trend in the moral acceptance of these behaviors by young people compared to ten years ago, the developing attitude of Slovak youth toward these issues might also be mirrored in the current political situation and general elections that took place on 30 September 2023. According to the survey of the AKO agency, one-fifth of voters decided to vote for the KDH (Christian Democratic Movement) political party and consider the PS (Progressive Slovakia) political party their second choice (Katuška 2023). This is interesting because the two parties are direct opposites regarding their value orientations. While KDH is known for and presents itself as a clear option for voters looking for protection of conservative Christian values, one of the main political priorities of PS is an effort to achieve a legislative basis for 'life partnerships' (equivalent of registered partnerships of same-sex couples). The party is also known for being most attractive to first-time voters (the age of voting in Slovakia is legally established to be 18 years old). Therefore, the fact that moral acceptance of phenomena such as the use of contraceptives and premarital sexual cohabitation is on the rise among young Slovaks is something churches should reflect on when communicating with young people. And while it is unlikely that young people of today will change their attitude towards these issues in the future (although it cannot be ruled out completely), sometimes they might feel that clergymen respond belatedly and incompletely to timely questions, which can cause problems for young believers (Ivanič 2019) or provide dissatisfactory information for a young religious person viewing the issues as a moral dilemma in a sense that they are looking a decision that should be consistent with their conscience and personal convictions (Chen et al. 2022). An example of that could be a recently published interview with the chairman of the Conference of Bishops of Slovakia in the online newspaper *Postoj*. The reporter began by noting that the secular environment often upbraids the Church for its commenting on mainly culturally ethical topics like the protection of life from the moment of conception to the moment of a natural death, or the marriage of a man and a woman. The reporter then asked whether the chairman thought that the Church should talk more about issues such as corruption, the environment, or the war on Ukraine (Rábara 2023). The chairman then provided the reporter with a quite voluminous and vague answer instead of a concrete and clear one. Additionally, data provided by the research of Štefaňak indicate that there are changes occurring in the views among Slovak Christian youth concerning some of the dogmatic points of Christian teachings. Štefaňak carried out religion-related research among Slovak youth in 2006 and 2016. The respondents included 708 people 17–18 years of age, localized in the northern parts of Slovakia, which are generally considered the most religious of Slovak regions. In 2006, 88.2% of his respondents identified with the Christian religion; in 2016 it was 89.7%. His research detected a decrease in the certainty of young Slovaks when it comes to some dogmatic truths of the Christian religion. Within the ten years, there were fewer of those who strongly believe that God is personal, that God became man and died for the people, that the Holy Testament is truly the word of God, or that people resurrect both with body and soul after death (Štefaňak 2019). Young people in Slovakia seem to

be gradually distancing themselves from many points of Christian teachings traditionally present in Slovakia. Thus, we might consider the future development of religious attitudes of young Slovaks and whether the trend might be taking place that was described by Kehl about Germany in the 1990s, when he claimed that German people were going to continue to claim their belonging to official confession yet form and believe in their own, individual doctrines (Kehl 2000). Štefaňak (2020b), on the other hand, considers the possibility of an increase in the number of young believers that do not claim to belong to any official confession or church. Among these youth, a small portion surely remains present that prefers returning to traditions in their religious life, sometimes even in the form of rejecting tolerance or cooperation with others, calling for the restoration of old traditions and the return of the Church to political life (Hetényi 2019).

Finally, we need to state that the interpretative value of some of the data from our survey is limited by the fact that some religiosity-related terms people commonly use come with a degree of subjectivity in understanding their meaning. Therefore, there might be some differences in what, specifically, respondents have in mind when reading, for example, the terms 'faith' and 'confession'.

## 4. Methods and Materials

The methodological starting point of this paper was a questionnaire focused on various religion and religiosity-related issues. The questionnaire was used within a survey that was part of a research project, APVV-17-0158, funded by APVV (Slovak Research and Development Agency) and distributed using the services of data collecting company Data Collect s. r. o. The basic sample consisted of the whole Slovak population. A selection pool of 1000 respondents of various sexes, ages, residences, and education from all Slovak regions was drawn from the basic set using a quota system. The obtained data were processed using correlation; statistical analysis focused on selected variables defined in theories and analyses of the decades studied. The procedures used were quantitative. As a result, the data presented in this article are statistical, and do not possess nor provide more information on cases whenever respondents chose the option "other". Also, because religion and faith-related terminology are often exposed to certain subjectivity in the interpretation of meaning, and there are differences in what specific respondents mean by, for example, the terms 'faith' or 'confession', we took extra caution in this regard. We asked respondents how religious they were/are when we wanted the data on their religiosity, and we provided them with the options "believing," "strongly believing," "not believing," etc., giving us data on the actual state of their faith. We asked them to what "church/religious institution/confession" they belong to when we wanted the data on their official religious affiliation. Despite this, it cannot be completely guaranteed that some respondents did not apply their subjectivity on these terms and did not, for example, mean that they do not identify with any official religious institution when they chose the option that they are "not believing".

## 5. Conclusions

Based on our data, there are probably more young people in Slovakia whose religiosity has weakened than those whose religiosity has strengthened. A large number of young people, but not larger than the number whose religiosity has weakened, do not reflect any change in their religiosity over the last ten years. About one-tenth is unable to make a judgment on that issue. Our research detected that reasons of a more rational (information-based) nature were prominent in weakening religiosity while reasons of a rather emotional nature were influential in its strengthening.

While it is generally agreed that today the media acts as one of the prominent information sources for Generation Z members, our research confirmed that the religious media in Slovakia does not satisfy this trend. Although almost 80% of young respondents in our research claimed affiliation with one of the big Christian churches in Slovakia, around 70% of them pay no attention to religious newspapers, radio, television, or websites. Most of the

other 30% only follow these religious media occasionally. This indicates that most young people in Slovakia do not seek religious-related information in religious media, and that religious media, therefore, plays an insignificant role in forming their religiosity, including its weakening or strengthening.

When respondents in our research reflected on the causes of the weakening of their religiosity, most frequently they chose the options of personal outlook and churches' behavior and scandals. This implies that young people in Slovakia perceive the Church as their social group on which they apply their critical thinking to the extent that the moral discredit of that church bears the potential to weaken their religiosity. Regarding moral issues, our research also detected the dissonance between the number of respondents affiliating with Christian churches in Slovakia and the respondents' attitudes toward cultural and ethical issues, namely premarital sexual relationships, the use of contraceptives, induced abortion, and homosexuality. The data imply that, except for induced abortion, at least half of young people in Slovakia morally accept these behaviors that are considered sins by Christian churches, and only around 14–23% of young people morally disapprove of them (even though almost 80% affiliate with one of these churches).

Overall, this paper lays out the possibility of a similar trend taking place among young people in Slovakia that Kehl described for Germany in the 1990s, that is, a trend of continuing claims of affiliation with the traditional churches while believing in and acting on individual religious and moral doctrines.

**Author Contributions:** Conceptualization, E.Ď. and P.K.; methodology, E.Ď.; validation, P.K.; formal analysis, P.K.; investigation, E.Ď.; resources, P.K.; data curation, P.K.; writing—original draft preparation, P.K.; writing—review and editing, E.Ď.; visualization, P.K.; supervision, E.Ď.; project administration, P.K.; funding acquisition, E.Ď. All authors have read and agreed to the published version of the manuscript.

**Funding:** The paper is the output of the project APVV-22-0204 Religiosity and Sustainability Values. Project is supported by Slovak Research and Development Agency.

**Data Availability Statement:** Data are available here: https://www.ukm.ff.ukf.sk/wp-content/uploads/2022/09/TABS-VIERA-2020-10.xlsx (accessed on 1 September 2023).

**Conflicts of Interest:** The authors declare no conflict of interest. The funders had no role in the design of the study; in the collection, analyses, or interpretation of data; in the writing of the manuscript; or in the decision to publish the results.

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
