# Peer review of "The Characteristics of the Religiosity of Youth in Slovakia Today"

_religions, doi:10.3390/rel14111433_

Round 1

Reviewer 1 Report

Comments and Suggestions for Authors

The analysis presented by the author is interesting and important as it provides more insight into the younger generation's relationship with the media and the influence of the media on them in Slovakia. The article in fact discusses the results of a quantitative statistical survey, however in order to foster a greater understanding of both the results and the context that is the subject of the analysis I would suggest that the author makes some additions.

- I think it is important to present in more detail the context in which the analysis is conducted. We know that the interviews are conducted in Slovakia, probably in recent times (after the outbreak of the pandemic) and that Slovakia is dominated by Christianity. What does this mean? What are the percentages in which the Christian religion is present in Slovakia and what are the other religious minorities? The author says that the Christian religion also "dominates" the media. What are the other areas in which this prevalence emerges? What role has church and religion historically played in the country and in the relationship with the state and civil society. These are some questions that I think would also help to better understand the results presented in the article. It also seems to me that the author makes "tentative" of mentioning  them in some parts. Here are some examples: p. 6 "The negative image of the church among adolescents and new adults can also be associated with the fact that they perceive the church as an institution that tries to intervene in the process of their personal decision-making."; p. 14 "As part of school teaching, the subject of media education is introduced into the curriculum. [..]".  I suggest to develop more these parts.

- the author begins the  article by talking about the importance that media and information have on people's lives, especially young people. I would suggest saying more about the relationship between religion and digital media. A reference could be the studies conducted by H. A. Campbell.

- Some personal observations, in my opinion, need to be scientifically substantiated:

p. 10, "What others think is right is also right for respondents." Why does the author trace respondents' answers back to the fact that there are universally accepted rules that apply to everyone? Because of age of respondents and of media role? The second  survey analyzed  tells us that the percentage of people who relate to mass media with religious content is low, does this mean that religious content also affects media that are not properly religious?

These are just vague observations, but I think the topic could be explored more.

- what is the difference between “faith” and “confession” that the author is thinking of when he refers to them on p. 13?

- p. 14: the author talks about conspiracy theory and alternative authority coming from the media-I would suggest saying more about it in relation to the context of Slovakia. 

- I would suggest giving some more information about the project and moving the part about the methodology used to the beginning of the article. Also, maybe this is information that has escaped to my attention, but in what years was the analysis conducted?

 - p. 13, the author says that "The basic ensemble consisted of the entire Slovak population. A selection pool of respondents of various sexes, ages, residence...," but at the beginning it says that the respondents were between 18 and 24 years old....perhaps this point should be better clarified.

I attach the file where in yellow are marked repetitions that I do not understand and that I think are typos.

Comments on the Quality of English Language

Author Response

Dear reviewer. Thank you for your valuable advice and comments. We tried to incorporate them as much as possible into the updated version of the work.

Reviewer 2 Report

Comments and Suggestions for Authors

1. Abstract needs to be revised for more clarity and understanding.

2.    Introduction desires improvement.

3.    The Study lacks a literature review, which is the backbone to find out the research gap in the existing literature on the study. 

4.    The study needs more academic references and sources he use only 12 sources which is not sufficient for that paper.

5.    The paper needs also the specification of Youth age and also the type of Media (Social, Electronic or Print)

6.    Research methodology also requires revision for more clarity.

7.    Overall language is suitable but I think needs revision for more understanding.

8.    Some important recommendations should be drawn separately.

9.    The reference style also wants correction, especially in the end notes.  

     10.  The Title of the study might be improved with decimation and mentioning the research approach.

Comments on the Quality of English Language

Overall language is suitable but I think the study needs revision  and proofreading for more understanding.

Author Response

(The authors gave the same response as above.)

Reviewer 3 Report

Comments and Suggestions for Authors

The article provides sufficient information. In some places, there is little lack of coherence which should be made up. In the abstract section, short implication of the study should be included.

Comments on the Quality of English Language

Minor editing is needed for improving the quality of the paper.

Author Response

(The authors gave the same response as above.)

Reviewer 4 Report

Comments and Suggestions for Authors

The starting point of the article is the thesis: The influence of the media on individuals  and society is undeniable. This thesis, however rather obvious, provides a broader perspective for the main research questions presented in the article. These questions relate to the importance of the media in shaping the views of religious young people. The author of the article formulates two hypotheses, which he seeks to prove. The first: Young believers critically use the media as a source of information for their own decision making on matters of faith; the second: Young believers prefer religious media as a source of information for their own decisions. The author analyses the theses on the basis of the results of the KEGA project research on the topic: Youth in a media-saturated society. In his view, the first thesis has been confirmed and the second not entirely. The author points out that the results of the studies on which he verifies the hypotheses are not sufficient to be properly interpreted. He points to the need for further research to verify the hypotheses adopted. Here, the question arises as to whether the hypotheses posed here can be verified on the basis of the research of the KEGA project? It seems that the author of the study himself has such doubts. Nevertheless, the attempt made here is largely successful and represents a first step towards further analyses based not only on empirical quantitative research, but above all on qualitative research. The theses presented in the study cannot be completely verified on the basis of quantitative research.

I therefore suggest before publishing the article:

-          point out in the introduction this difficulty in interpreting the research results

-          expand the range of literature for commenting on research results

-          clarify what is meant by KEGA project research (abbreviation KEGA)

-          indicate whether there is access to the full research results of the KEGA project: Youth in a media-saturated society

Summary: The manuscript is clear and relevant to the study of pastoral theology. The sources cited would need to be expanded. The design is adequate to test the hypothesis, albeit not fully. The tables are adequate and present the data appropriately. They assist in the interpretation and understanding of the data. The article is original and shows the author's research independence.

Author Response

(The authors gave the same response as above.)

Reviewer 5 Report

Comments and Suggestions for Authors

This reviewer has several significant concerns regarding this submission. The very title of the paper illustrates at least two of these concerns. First of all, it suggests that the focus is broadly on the importance of media in forming opinions among religious youth. Yet the paper is focused specifically on Slovakian youth. This needs to be stated in the title. Secondly, the title is also misleading because the study focuses not only on religious youth but also non-religious youths. The paper is actually about “forming religious opinions among Slovakian youth.” Please note that a simple change of title is not sufficient here. The concerns illustrated by the title are generic to the entire paper which needs to be revised accordingly.

Another major concern is widespread practice by the author(s) of making unsubstantiated assumptions.  Any statement in this paper which uses the verb “assume” should probably be deleted entirely. On page 5, in fact, the author(s) make the claim that something is “indisputable” when they have not proof of this and cannot make that claim. Also “we can say” at 454 is another form of assumption. Dealing with theses issues will require major rethinking and rewriting of this submission.

This reviewer’s major concern, however, is that the authors do not provide convincing evidence based upon this concern that ANY of the respondents are influenced by media in the formation of any opinions, including religious ones. The response “information, outlook, experience” could come from a variety of sources, including the media, but not necessarily significantly. The wording of this survey was not designed in such a way to provide the kind of data the authors wanted from this survey. See especially lines 180ff. This reviewer, therefore, questions the validity of the entire paper.

Furthermore, there are many places which require grammatical editing or clarification. This reviewer offers only a few examples here:

1.       The authors are inconsistent with their use of personal pronouns like “he,” “she” and “they.” The authors need to adapt one pronoun to refer to respondent(s) and use that one consistently.

2.       Page 1: Reference to KEGA project Youth needs explanation.

3.       Page 3, line 91: “question of: Which”. This is very poor English. Try “question of determining which”

4.       Page 3, line 113: “handling it childhood” There is something wrong here which suggests that the paper has not been carefully edited.

5.       Page 3 line 132: use of word “complement” does not make sense to this reviewer, who has no idea what the author(s) intend to say here.

6.       Whenever the author(s) refer to response categories in the survey, these categories MUST be put in quotation marks. One example (of many): Pg. 6, line 212 “other causes” or “I cannot judge”

7.       Line 181: “outlook of experience” should be “outlook or experience” (another example of insufficient editing)

8.       Line 242: The word “childlike” would be much preferable to “infantile” (which would have very negative connotations in this context)

9.       Titles for figures should appear on same page with figure (see figure on page 9) and author(s) should not just refer to “the figure” but should always refer to it by name, e.g., line 324

CChart on pg 9: Note typo for “I don’t know”

1There are two Figure 4s on page 11!

1In both Figures 4 the response “at all” should be “not at all”

1Lines 481-482: The authors provide no evidence that the church and priests respond in this way.

1 Line 502: “conciliating” This reviewer has no idea what the author(s) mean here.

1 Line 504: “unbeliever” is preferable to “infidel” (which is a derogatory term)

Comments on the Quality of English Language

This reviewer has several significant concerns regarding this submission. The very title of the paper illustrates at least two of these concerns. First of all, it suggests that the focus is broadly on the importance of media in forming opinions among religious youth. Yet the paper is focused specifically on Slovakian youth. This needs to be stated in the title. Secondly, the title is also misleading because the study focuses not only on religious youth but also non-religious youths. The paper is actually about “forming religious opinions among Slovakian youth.” Please note that a simple change of title is not sufficient here. The concerns illustrated by the title are generic to the entire paper which needs to be revised accordingly.

Another major concern is widespread practice by the author(s) of making unsubstantiated assumptions.  Any statement in this paper which uses the verb “assume” should probably be deleted entirely. On page 5, in fact, the author(s) make the claim that something is “indisputable” when they have not proof of this and cannot make that claim. Also “we can say” at 454 is another form of assumption. Dealing with theses issues will require major rethinking and rewriting of this submission.

This reviewer’s major concern, however, is that the authors do not provide convincing evidence based upon this concern that ANY of the respondents are influenced by media in the formation of any opinions, including religious ones. The response “information, outlook, experience” could come from a variety of sources, including the media, but not necessarily significantly. The wording of this survey was not designed in such a way to provide the kind of data the authors wanted from this survey. See especially lines 180ff. This reviewer, therefore, questions the validity of the entire paper.

Furthermore, there are many places which require grammatical editing or clarification. This reviewer offers only a few examples here:

1.       The authors are inconsistent with their use of personal pronouns like “he,” “she” and “they.” The authors need to adapt one pronoun to refer to respondent(s) and use that one consistently.

2.       Page 1: Reference to KEGA project Youth needs explanation.

3.       Page 3, line 91: “question of: Which”. This is very poor English. Try “question of determining which”

4.       Page 3, line 113: “handling it childhood” There is something wrong here which suggests that the paper has not been carefully edited.

5.       Page 3 line 132: use of word “complement” does not make sense to this reviewer, who has no idea what the author(s) intend to say here.

6.       Whenever the author(s) refer to response categories in the survey, these categories MUST be put in quotation marks. One example (of many): Pg. 6, line 212 “other causes” or “I cannot judge”

7.       Line 181: “outlook of experience” should be “outlook or experience” (another example of insufficient editing)

8.       Line 242: The word “childlike” would be much preferable to “infantile” (which would have very negative connotations in this context)

9.       Titles for figures should appear on same page with figure (see figure on page 9) and author(s) should not just refer to “the figure” but should always refer to it by name, e.g., line 324

10.   Chart on pg 9: Note typo for “I don’t know”

11.   There are two Figure 4s on page 11!

12.   In both Figures 4 the response “at all” should be “not at all”

13.   Lines 481-482: The authors provide no evidence that the church and priests respond in this way.

14.   Line 502: “conciliating” This reviewer has no idea what the author(s) mean here.

15.   Line 504: “unbeliever” is preferable to “infidel” (which is a derogatory term)

Author Response

(The authors gave the same response as above.)

Round 2

Reviewer 5 Report

Comments and Suggestions for Authors

Other than revising the title of the paper and a few relatively minor editorial changes, the author(s) have not significantly addressed the serious and substantive concerns expressed in my earlier review. I will use my earlier comments as a basis for explaining my evaluation of this revised paper. My earlier comments are in plain font. New comments are in bold.

  This reviewer has several significant concerns regarding this submission. The very title of the paper illustrates at least two of these concerns. First of all, it suggests that the focus is broadly on the importance of media in forming opinions among religious youth. Yet the paper is focused specifically on Slovakian youth. This needs to be stated in the title. Secondly, the title is also misleading because the study focuses not only on religious youth but also non-religious youths. The paper is actually about “forming religious opinions among Slovakian youth.” While the author(s) have made appropriate changes to the title in the revised version, please note that the old title is still used on the submission form.

Please note that a simple change of title is not sufficient here. The concerns illustrated by the title are generic to the entire paper which needs to be revised accordingly. These concerns are elaborated below. I realize that four other readers have recommended publication after revision. I fear that I cannot follow their recommendation. At this point I consider this paper so flawed in terms of methodology and conclusions. These flaws are so substantive that no revisions other than complete rewriting could address my concerns.

Another major concern is widespread practice by the author(s) of making unsubstantiated assumptions.  Any statement in this paper which uses the verb “assume” should probably be deleted entirely. On page 5, in fact, the author(s) make the claim that something is “indisputable” when they have not proof of this and cannot make that claim. Also “we can say” at 454 is another form of assumption. Dealing with these issues will require major rethinking and rewriting of this submission. The authors have made no effort in the revision to address my concern about their frequent use of broad assumptions without the support of evidence. The word “indisputable” still appears, at line 243 in the revised version. “we can say” still appears, not at line 514 in the revised version. No evidence is provided for these statements which may or may not be true. There are many more instances of this in the paper.

This reviewer’s major concern, however, is that the authors do not provide evidence based upon this concern that ANY of the respondents are influenced by media in the formation of any opinions, including religious ones. The response “information, outlook, experience” could come from a variety of sources, including the media, but not necessarily significantly. The wording of this survey was not designed in such a way to provide the kind of data the authors wanted from this survey. See especially lines 180ff. This reviewer, therefore, questions the validity of the entire paper. This concern is significant. The author(s) consider “information, outlook, experience” to be media-based, but there is no evidence for this. As far as I can tell, the original survey was not designed in a way to provide this evidence. At lines 566-7 the author(s) themselves admit the need for a more focused survey.

Comments on the Quality of English Language

Furthermore, there are many places which require grammatical editing or clarification. This reviewer offers only a few examples here:

1.       The authors are inconsistent with their use of personal pronouns like “he,” “she” and “they.” The authors need to adapt one pronoun to refer to respondent(s) and use that one consistently. This concern has not been addressed. In fact, editorial changes have made the inconsistency more widespread. I would suggest that the authors use only third person plural pronouns and adjectives (“they”, “them”, “their”). I would also recommend replacing “man” at line75 with ‘humans” or “human beings.”

2.       Page 1: Reference to KEGA project Youth needs explanation. This clarification has been provided. But not consistently. At line 32 the authors write “Youth in  a Media-Saturated Society” but at line 46 they write “You in a media-saturated society.’ Be consistent in use of capitalization!

3.       Page 3, line 91 (line 152 in revised version): “question of: Which”. This is very poor English. Try “question of determining which” The authors tried to address this concern by simply changing “which” to “what” but this does not really address the awarkness of this sentence. Here is better wording:  “We are faced with the question of determining what are the main ways that young believers seek answers….”

4.       Page 3, line 113: “handling it childhood” There is something wrong here which suggests that the paper has not been carefully edited. In the revision at line 173ff he author(s) tried to address this concern by replacing “childhood” with “using” and deleting a repetitious following sentence. But the result is ungrammatical: see “processing using it” at lines 173-74.

5.       Page 3 line 132: use of word “complement” does not make sense to this reviewer, who has no idea what the author(s) intend to say here. Good edit here.

6.       Whenever the author(s) refer to response categories in the survey, these categories MUST be put in quotation marks. One example (of many): Pg. 6, line 212 “other causes” or “I cannot judge”. The author(s) have made some effort to follow this suggestion but there are still many places where quotation marks are lacking.

7.       Line 181: “outlook of experience” should be “outlook or experience” (another example of insufficient editing) done

8.       Line 242: The word “childlike” would be much preferable to “infantile” (which would have very negative connotations in this context) done

9.       Titles for figures should appear on same page with figure (see figure on page 9) and author(s) should not just refer to “the figure” but should always refer to it by name, e.g., line 324. This remains a significant formatting problem which I trust will be addressed before the paper were published (over my obmection).

CChart on pg 9: Note typo for “I don’t know”. This is figure 3 on page 10. The author(s) tried to address this but failed to put a space between “I” and “don’t”.

 There are two Figure 4s on page 11! This problem has been addressed.

1In both Figures 4 the response “at all” should be “not at all” This problem has been addressed.

1Lines 481-482: The authors provide no evidence that the church and priests respond in this way. The author(s) added a reference to support this statement, but I think that some elaboration is warranted in the paper.

1 Line 502: “conciliating” This reviewer has no idea what the author(s) mean here. Satisfactory edit made.

1 Line 504: “unbeliever” is preferable to “infidel” (which is a derogatory term). Not only have the author(s) ignored this concern, but they have added additional uses of the word, e.g., at line 544. The word “infidel” is usually used in English to refer to Christian vs. Moslem in an historical context. A much less “loaded” and controversial word for this paper would be “unbeliever” or perhaps “former believer.”

After reading the revision, here are additional concerns that have arisen for me.

The pie charts often use different tones of the same color so that it is often difficult to distinguish categories.

In Figure 1, the percentage is not provided for one of the blue categories.

In Figure 1, one of the categories is labeled “aging.” In English this word usually prefers to the “aging” experienced by the elderly. Perhaps “growing older” or “maturity” would be better here.

In figure 2, the percentage in the darkest blue section is not visible due to the darkness of the color.

In fact it would be helpful if the percentages were added in parentheses after the category. E.g., in Figure 3, change “Other” to “Other (78.20%)”

Line 47: “We’re”. Use of contractions is not appropriate in formal, scholarly English.

Line 116: “not only” These words should always be followed by “but also” but no “but also” ever appears. There are several examples of this poor English usage.

Line 187: “That is.” Poor grammar. Better English is “…in the past; that is, there has been…”.

Lines 191ff: This reviewer would like to see an additional chart showing how young people’s attitudes towards religion have changed. It is also important to provide statistics on the actual number of respondents. How many youth said their faith had decreased compared to the number of youth whose faith had remained the same or increased? The Proof of Theorem 1 at line 345 depends, to a certain extent, on the actual numbers of these two groups. The author(s) argue that “we can say that those young believers use the media as a source of information for their religious faith” but the data provided in the paper suggests that this is not necessarily true for the youth whose faith remains strong. If they are a larger group that the youth whose faith has weakened then the author(s) cannot say, without qualification, that “young believers use the media….” . The authors add a citation to Stefanak 2020a to support this statement, but, under the circumstances, this needs more than a parenthetical. Provide specific data or quote Stefanak providing such data.

Also for Figures 4 and 5 it would be important to show, separately, the responses of youth believers and youth non-believers. The percentages would be very different for each category!

Lines 222-226: Some of these categories remain unexplained and do not appear in any pie chart. For example “Velvet Revolution” “the change in the situation”

Line 463: “how often they watch religious periodicals”. One does not “watch” a periodical. Rather it id “read” whether on paper or on line!

Line 529: “I mature with the media” This sentence is incomprehensible. 

Author Response

Dear reviewer, we are sending a revised article in the attachment. We also requested to change the name of the article in the system.
